# Measuring quality of life in trials including patients on dialysis: how are transplants and mortality incorporated into the analysis? A systematic review protocol

Hannah M Worboys  ,[1] Nicola J Cooper,[1] James O Burton,[2] Laura J Gray[1]

[1]Department of Health Sciences, University of Leicester, Leicester, UK
[2]Department of Cardiovascular Sciences, University of Leicester, Leicester, UK

**Correspondence to**
Hannah M Worboys;
hw315@leicester.ac.uk

## ABSTRACT

**Introduction** It is estimated that 25 000 people in the UK receive dialysis. Dialysis is an intrusive and time-consuming intervention that causes significant reductions in quality of life. When enrolled in a clinical trial, often some patients drop out of the study either because they die, receive a kidney transplant or are lost to follow-up for other reasons. It is unclear how these events are dealt with when analysing quality of life measures within clinical trials. This review will assess current practice for dealing with loss to follow-up in trials including patients on haemodialysis. The methods currently used will be analysed in terms of their adequacy and will form the basis of future work assessing the most appropriate methods to employ under these circumstances. The results of this review will feed into recommendations for future nephrology trials.

**Methods and analysis** A systematic search of electronic databases including MEDLINE and the Cochrane Library will be conducted to find clinical trials enrolling patients on haemodialysis that measure quality of life using either the kidney disease quality of life (KDQoL) or the short form 36 health survey (SF-36) (or any variation of these two measures). Ongoing trials will be identified through a search of trial registers. Articles will be screened against inclusion/exclusion criteria and data will be extracted using a predetermined data extraction form. General information such as the title, location, trial design will be extracted along with more specific information on how the study dealt with patients that died or received a transplant before the end of the follow-up period. Two independent reviewers will perform screening and extraction. Disagreements will be resolved by discussion or by a third independent reviewer. Data synthesis will be performed as a narrative summary.

**Ethics and dissemination** Ethics approval is not required. Dissemination will be by publication in a peer-reviewed journal.

**PROSPERO registration number** CRD42020223869.

### Strengths and limitations of this study

► This review aims to collate the methods employed for dealing with specific missing outcome data, patients who dropout due to transplants, adverse effects and death before the end of the follow-up period.
► The Preferred Reported Items for Systematic Reviews and Meta-Analyses guidelines will be adhered to.
► A comprehensive search strategy, peer reviewed by a clinician, librarian and methodologists, will be undertaken in several electronic databases of peer-reviewed journals and trial registers.
► It is acknowledged that chronic kidney disease has high prevalence across the world, however, including only English language publications is not expected to have a significant effect on the results of this review.
► It is expected that many studies will have included insufficient information about the conduct of their data analysis, authors will be contacted which may delay the review but will help resolve this issue.

## INTRODUCTION

The number of patients suffering from chronic kidney disease (CKD) across the world is increasing. Advanced cases of CKD require dialysis to replace the function of the kidneys. It is approximated that there are 25 000 dialysis patients in the UK.[1] Dialysis is costly and requires many healthcare resources including hospital equipment, nurse and physician time. It is also an intrusive procedure and requires multiple trips to a clinic every week. This imposes a significant burden on patients and their families. Patients are often limited in their ability to work and care for dependents, as well as experiencing headaches, fatigue and other side effects as a result of dialysis.[2] These factors significantly affect both adherence to the treatment and quality of life (QoL). On average, one in three patients with CKD experiences depression[3] and non-adherence to treatment is escalating levels of patient mortality.[4] The adverse effects on the well-being of patients with CKD are well known; this has led to a large increase

in clinical trials that aim to improve QoL. Despite a large number of trials, QoL among patients on dialysis remains low. Many trials have failed to reflect important improvements in outcomes and hence have not lead to changes practice. This may be linked to limitations of the trial design such as small sample size, inability to blind participants, using QoL measures not validated in the dialysis patient population and dealing with missing data inadequately.[5]

The problems relating to recruitment and retention to dialysis trials are well-documented in the literature.[6] Strategies to improve recruitment into kidney disease trials have been developed and include improving education, increasing information prior to potential trials and incentivising participation.[7] As well as this, core outcome sets such as those developed by the Standardised Outcomes in Nephrology for Haemodialysis[8] (SONG-HD) collaboration are designed to ensure the most important outcomes are incorporated in trial design. Initiatives such as SONG-HD has led to the recognition of the importance for condition-specific patient reported outcome measures (PROMs). The kidney disease quality of life (KDQoL) questionnaire is a condition-specific QoL measure that is designed to measure health-related QoL (HRQoL) issues associated with patients with CKD.[9] The KDQoL-36 is a short form of the KDQoL that includes the 12-item short form survey (SF-12), as well as questions relating to; the burden of kidney disease; symptoms and problems of kidney disease; and the effects of kidney disease. The KDQoL-36 scores highly in consistency, validity and reliability[10] and is a highly used PROM among dialysis patients. The KDQoL is used internationally and has been translated into over 20 languages.[11] The short form 36 health survey (SF-36) is a generic measure of health that has also been used widely in trials, including measuring QoL among patients with CKD.

Despite many researchers developing appropriate ways to deal with missing data, little is mentioned about the specific types of missing data that occur in dialysis trials. During trials in dialysis, a number of patients will either receive a kidney transplant or die before the intended data collection point. For example, the PIVOTAL trial,[12] had 515 deaths (24% of patients randomised) and 371 transplants (17% of patients randomised) and had a median follow-up of 2.1 years. Outcomes for patients who have a transplant may or may not be collected. For those who have died, outcomes may be set to an arbitrary value or to missing in the analysis. Therefore these two events may result in missing values. It is unclear how this missing data has been dealt with in the data analysis of previous trials. Intention-to-treat (ITT) analysis is the gold standard for randomised controlled trials (RCTs) and involves analysing all patients that were randomised into a trial irrespective of the subsequent events. ITT is the preferred approach as it reduces the risk of bias, maintains randomisation and better reflects the reality of receiving treatment in the real world. Under ITT, it is therefore inappropriate to exclude missing outcome data that occurs due to patients who receive a transplant or die prior to the follow-up period. Excluding these patients may lead to overestimating/underestimating the intervention effect. It is important to find an appropriate method to account for these two events, such that an accurate estimation of the intervention effect can take place.

## Objectives

This systematic review aims to evaluate current practice for dealing with missing outcome data due to patient mortality, kidney transplantation and other types of drop out. The review will appraise trials that use the KDQoL or SF-36 when measuring QoL. We focus on the KDQoL and SF-36 as these are commonly used in trials of dialysis patients;[13] however, the findings can be expanded to all QoL measures.

The proposed systematic review will answer the following questions.

1. Do trials including dialysis patients measuring QoL report how they deal with patients who die, have a transplant or are lost to follow-up?
2. What methods are currently used for dealing with patients who die, have a transplant or are lost to follow-up up in the analysis of trials?
3. How is the KDQoL reported and what statistical methods are used to analyse the QoL of patients?

## METHODS

The systematic review will follow the Preferred Reporting Items for Systematic Reviews and Meta-Analyses (PRISMA) guidelines for reporting the results of systematic reviews.[14] The PRISMA Protocol checklist for systematic review protocols was used when writing this protocol[15] and is provided in the online supplemental information 1.

### Peer-reviewed literature

The search strategy will be developed by the primary author with the assistance of a specialist health sciences librarian and peer-reviewed by an Honorary Consultant Nephrologist at the Leicester General Hospital. The electronic databases MEDLINE, Web of Science, Cochrane Central Register of Controlled Trials, Scopus and Cumulative Index of Nursing and Allied Health Literature will be searched using combinations of keywords and topics. The main search strategy, written for MEDLINE, is included in the online supplemental information 1 and will be adapted for use in subsequent databases. Full search strategies will be included in the Appendix of the review. The databases will be searched from inception to 31 December 2020. Searches will be re-run before the final analysis, any studies published in the intervening period will be retrieved and included. The search will be limited to publications readily available in the English language. Search results will be exported and sorted using EndNote V.X9. All studies included will have their reference lists checked for additional studies.

## Ongoing studies

Trials registers are an important source for identifying additional randomised trials. ClinicalTrials.gov, the International Clinical Trials Register Search Portal, the European Union Clinical Trials Register and the International Standard Randomised Controlled Trial Number will be searched to identify ongoing studies for inclusion in the review. Searches will be extensive to reduce the risk of publication bias.

## Inclusion/exclusion criteria

For inclusion, the trial must be testing an intervention within a population of haemodialysis patients. The trial must measure QoL as an outcome. Trials must use the KDQoL (any variation) or the SF-36 (any variation) to measure QoL. Trials considered for inclusion will be phase 3 clinical trials of any design. Trials included will have recruited adult populations (18+) as the KDQoL and SF-36 are unlikely to be asked directly to children. There is no restriction on the setting or location of the trial. Articles must be either reported or readily available in the English language.

## Screening

Titles, authors, abstracts, the date and location of publication will be exported into EndNote V.X9. Duplicates will be removed. Titles and abstracts will be screened for inclusion. The full-text articles will be screened for confirmation of suitability if the abstracts do not have sufficient information. Authors will be contacted if after full-text screening it is still not clear whether to include/exclude an article. All articles will be screened independently by two reviewers. Any disagreement on inclusion will be resolved by discussion or with another reviewer. A characteristics of excluded studies table will be presented in the Appendix of the review, detailing reasons for exclusion.

## Data extraction

After screening, data extraction will be performed using a predetermined extraction form. Two data extraction forms will exist; one focusing on the approach used in completed trials; the other focusing on the planned approach for ongoing trials. Two reviewers will do the extraction, with differences discussed. A third reviewer will resolve if necessary. Pilot data extraction will test the forms prior to the main analysis.

Data to be extracted includes:

General
- ► Title.
- ► Lead Author.
- ► Year.
- ► Author correspondence.
- ► Study design.
- ► Equivalent size control group.
- ► Country.
- ► Multicentre (Y/N).
- ► Population.
- ► Type of dialysis.

- ► Research ethics obtained.
- ► Informed consent obtained.
- ► Inclusion of Consolidated Standards of Reporting Trials PRIMSA flow diagram.
- ► Number randomised.
- ► Number analysed.
- ► Number of dropouts, overall and by reason.
  Mortality.
  Transplant.
  Other reasons, for example, due to an adverse event.
- ► Time points of data collection.
  HRQoL specific
- ► HRQoL instrument used.
- ► HRQoL primary outcome (Y/N).
- ► Methods used to account for missing HRQoL data.
- ► KDQoL reporting methods.
- ► KDQoL primary analysis method.
  Additional information for ongoing trials
- ► Status of trial.
- ► Anticipated sample size.
- ► Planned approach to main analysis (ITT, complete case, per protocol).
- ► Trial protocol available (Y/N).

## Outcomes

- ► The number of studies that mention how death or transplants are dealt with in the analysis.
- ► Methods used to account for deaths and transplants in the analysis of QoL.
- ► Most common methods used.
- ► Number of studies that fail to mention how they account for deaths and transplants in the analysis of QoL.
- ► Number of studies where death or transplants account for more than 5% of patients randomised.
- ► Associations between methods and the risk of bias.

## Risk of bias assessment

RCTs are the gold standard of evidence-based medicine. However, they are open to bias if poorly designed or executed. When conducting a systematic review, in order to make reliable conclusions, the reviewers must consider the quality of the studies included. The Cochrane collaboration's tool for assessing the risk of bias (RoB)[16] has been used in a significant number of previous systematic reviews. An updated version of this tool, the RoB V.2.0[17] will be used in this review to assess the RoB for each trial. Judgements about the RoB will be made independently by two reviewers and disagreements resolved through discussion.

There are five domains of bias, each with several elements. Each domain is concluded as 'low', 'high' or 'some concerning' level of risk. Completion of all domains leads to an overall risk of bias judgement. The aim of the quality assessment is not to further exclude studies from the review but instead to find the strengths and limitations of each study.

The sources of bias to be assessed are

- ► Domain 1: RoB arising from the randomisation process.
- ► Domain 2: RoB due to deviations from the intended interventions.
- ► Domain 3. RoB due to missing outcome data.
- ► Domain 4: RoB in measurement of the outcome.
- ► Domain 5: RoB in the selection of the reported result.

For non-randomised controlled trials, the Cochrane RoB for non-randomised studies (ROBINS-I)[18] will be used to assess the quality of the trial. Similar to the RoB tool, the ROBINS-I will lead to an overall RoB judgement for each trial.

### Patient and public involvement

No patient involved. It was not appropriate or possible to involve patients or the public in the design, or conduct, or reporting, or dissemination plans of our research.

### Data synthesis
#### Completed trials

Completed trials that meet the inclusion criteria will be analysed using either RoB or ROBINS-I tool. The key information from the extracted data, alongside the RoB will be tabulated. Articles will be grouped into whether they have stated the methods used to deal with death and transplant patients. Articles that have will then be categorised into methods used. Authors of articles that have not will be contacted for this information. Responses will then be categorised into methods used or non-response. The results will be synthesised using descriptive statistics. Where possible, the percentage of patients who die or receive a transplant out of the total number of patients randomised will be calculated respectively for each trial. This review will highlight the common methods as well as the percentage of studies that fail to mention this issue entirely. Associations between reporting and the RoB will be assessed. The results of the review will be interpreted in line with the limitations (RoB) of each study.

#### Ongoing trials

The key information from the extracted data will be tabulated. Articles will be grouped into whether they have anticipated the methods they will use to deal with death and transplant patients. Articles that have will then be categorised into methods used. Authors of articles that have not will be contacted for this information.

Extensive systematic searches will strengthen the findings of this review. It is anticipated that the results of this review will identify the methods used to deal with patients who die or receive a transplant in the analysis of QoL. It is also anticipated that this review will identify that the methods used are inconsistent and the need to develop an acceptable method to be used consistently in nephrology trials.

### DISCUSSION

Previous papers highlight the general poor methodological quality of nephrology trials.[19] As well as this, papers have shown there is a mishandling of missing QoL data

in a range of clinical areas.[20] A scoping review showed that there is potentially a high prevalence of trials of dialysis patients that do not include adequate methods, this review aims to show definitively how high the prevalence is.

The scoping review indicated some use of complete case analysis (CCA) in this discipline and the literature shows wide use of CCA in other clinical areas.[21] This review will assess whether this is the case for nephrology trials, where dealing with QoL data due to transplants is a unique issue. The limitations of CCA are that if patients have died or have dropped out due to adverse effects from the treatment, excluding them from the analysis could lead to an overestimation of the intervention and QoL estimates. Methods of imputation have also been implemented[20]; however, this is only justified if the missing data mechanisms are valid. If patients drop out due to the treatment, the assumption of the data being missing completely at random is invalid, and imputation methods cannot be employed. This review hopes to find other methods that have been used; if not, this review will highlight the need for more sophisticated methods to be introduced into the data analysis of clinical trials.

The information provided in this review will form the basis of further investigation into the most appropriate statistical methods of dealing with missing QoL data due to deaths, transplants and other reasons for drop out. Simply excluding these patients from the analysis in favour of CCA may lead to biased results, but anecdotal evidence suggests this maybe the most common approach used. Although the focus of this review is on kidney disease, the results will be applicable to any trials which include QoL as an outcome.

**Contributors** HMW, LJG, NJC and JOB were involved in all stages of writing the systematic review protocol and contributed to the concept and design, drafting, revision and approval of the final version. The search strategy for peer-reviewed literature and trial registers was created by HMW and reviewed by JOB and LJG.

**Funding** NIHR Applied Research Collaboration East Midlands. HMW is funded by the National Institute for Health Research (NIHR) Applied Research Collaboration East Midlands (ARC EM). The views expressed are those of the author(s) and not necessarily those of the NIHR or the Department of Health and Social Care.

**Competing interests** None declared.

**Patient consent for publication** Not required.

**Provenance and peer review** Not commissioned; externally peer reviewed.

ORCID iD
Hannah M Worboys http://orcid.org/0000-0003-3958-0180

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
