## [Reviewer comments · BMJ Open]

ARTICLE DETAILS

TITLE (PROVISIONAL)	Measuring quality of life in trials including patients on dialysis: How are transplants and mortality incorporated into the analysis? A systematic review protocol
AUTHORS	Worboys, Hannah; Cooper, Nicola; Burton, James; Gray, Laura

VERSION 1 – REVIEW

REVIEWER	Devika Nair Vanderbilt University
REVIEW RETURNED	27-Feb-2021

GENERAL COMMENTS	This is a protocol for a systematic review aimed to describe how transplants and mortality are incorporated into analyses of trials that measure quality of life among patients receiving dialysis. Prior systematic reviews have not yet addressed this. Overall, I would provide better justification for why the information provided by this review fills an important knowledge gap in the literature. Does this improve the rigor of current studies in this space? What is the estimated prevalence of existing trials that do not include methods for competing risk analysis, etc? I have the following Major and Minor comments intended to strengthen the authors' contribution: Major - Please justify why the two measures specified by the investigators are those which have been chosen. There are numerous measures of quality of life, and the KDQOL-36 has its own limitations. Though it is the most frequently used measure to measure quality of life in kidney disease, there is no universally agreed upon measure to assess quality of life in kidney disease. The Patient Reported Outcome Measurement Information System (PROMIS) quality of life measures are being increasingly used.- Please include examples of ways that kidney transplantation and death would be accounted for in these analyses (competing risk analysis, censoring, using a cause-specific approach) as well as the limitations of these approaches- Consider other including other censoring events, such as loss to follow-up- Is there evidence that trials measuring quality of life do NOT include these methods? Please cite evidence for this. Minor - Would specify in the abstract itself whether the investigators are including patients receiving hemodialysis, peritoneal dialysis, or both- Would provide better justification in the abstract itself rationale for why the information gained in this systematic review
--

REVIEWER	Sarbjit Jassal Toronto General Hospital, Medicine
REVIEW RETURNED	10-Mar-2021
GENERAL COMMENTS	Important work. I look forward to answers, though suspect there is no effective way to manage these biases.

VERSION 1 – AUTHOR RESPONSE

Dear Shona Reeves,

Thank you for giving us the opportunity to submit a revised draft of our manuscript titled “Measuring quality of life in trials including patients on dialysis: How are transplants and mortality incorporated into the analysis? A systematic review protocol”

The authors appreciate the time and effort dedicated by the reviewers to provide feedback on the manuscript. We have been able to incorporate changes to reflect most of the suggestions provided by the reviewers. We have tracked the changes within the manuscript. Here is a point-by-point response to the comments.

Provide better justification for why the information provided by this review fills an important knowledge gap in the literature.

The information provided in this review will form the basis of further investigation into the most appropriate methods of dealing with missing QoL data due to deaths, transplants and other types of drop out. Simply excluding these patients from the analysis in favour of complete case analysis (CCA) may lead to biased results, but anecdotal evidence suggests this may be the most common approach used. Although the focus of this review is on kidney disease, the results will be applicable to any trials which include quality of life as an outcome. This has been added to the discussion.

Much literature has investigated the methodological quality of nephrology trials and concluded that there are several issues limiting them. As well as this, many previous studies have agreed that missing QoL data is generally mishandled in the analysis of clinical trials. These are now cited in the paper.

Does this improve the rigor of current studies in this space?

We believe the results from this work will improve the rigor of current studies. This review is the first stage in a programme of work, the aim of this review is to show the scale of the mishandling of missing QoL data and find methods used in current practice to account for the data. The second part of the project is to compare statistical methods and make recommendations for future data analysis of clinical trials. This programme of work should therefore improve the analysis methods used to analyse quality of life data going forwards.

What is the estimated prevalence of existing trials that do not include methods for competing risk analysis, etc?

Based on an initial scoping review, it is anticipated there is a high prevalence of existing clinical trials that do not use methods other than CCA. The results from the review will show definitively how high this prevalence is. This has been added to the discussion.

Please justify why the two measures specified by the investigators are those which have been chosen.

HRQoL is a key outcome for dialysis patients¹. The KDQOL and SF-36 are the most widely used tools for assessment of this outcome². The KDQOL is one of the few disease-specific patient reported outcome measures for kidney disease and has incorporated important domains that are not included in generic measures. Limiting to these two measures also keeps the review a manageable size given the amount of quality of life research that goes into this patient population. However the findings can be extended to other QoL measures. This has been added to the objectives.

Please include examples of ways that kidney transplantation and death would be accounted for in these analyses as well as the limitations of these approaches.

Results from the scoping review show that CCA is the most widely used method. The limitations of this are that if patients have died/ dropped out from the study due to the intervention and are excluded from the analysis, the intervention effect and QoL estimate could be overestimated. As well as CCA, imputation has also been used. Imputation can only be implemented if the missing data mechanisms assumptions are valid. As missing QoL data such as deaths, adverse events and transplants are missing not at random (MNAR) some imputation methods are invalid. The review will give definitive data on how trials have dealt with these issues. This has been added to the discussion.

Consider other including other censoring events, such as loss to follow-up

Other censoring events such as adverse effects and discontinuation will be added to extraction. We have added this to the methods section.

Is there evidence that trials measuring quality of life do NOT include these methods? Please cite evidence for this.

In a review of 87 RCTs, randomly selected from a range of clinical disciplines, 50% used CCA. Multiple imputation was used but mostly in sensitivity analysis. The authors conclude there is a large gap between the methods relating to missing data and their use in applications³.

In a review of 285 RCTs published in 2005/6, the majority of studies did not impute missing data and carried out CCA⁴. This is now cited in the paper. The aim of this review is to assess the current practice specifically in nephrology trials, where dealing with QoL data due to transplants is a unique issue.

Would specify in the abstract itself whether the investigators are including patients receiving hemodialysis, peritoneal dialysis, or both.

This has been added to the abstract.

Would provide better justification in the abstract itself rationale for why the information gained in this systematic review

We have added a justification to the abstract.

Additional changes

We would also like to inform the reviewers that we have made changes to the data extraction section of the protocol to better reflect the current work. The KDQoL is inconsistently reported throughout studies, for example some report a kidney-disease component summary, some report other summary scores and some report "KDQoL total score" against the recommendations of the measure developers. This review is being extended to extract information on the way the KDQoL is reported and the primary data analysis for measuring QoL.

VERSION 2 – REVIEW

REVIEWER	Devika Nair Vanderbilt University
REVIEW RETURNED	22-Jul-2021
GENERAL COMMENTS	The authors have satisfactorily addressed my comments and suggestions.